# Comparison of Two Viscoelastic Testing Devices in a Parturient Population

**DOI:** 10.3390/jcm13030692

**Published:** 2024-01-25

**Authors:** Daniel Gruneberg, Stefan Hofer, Herbert Schöchl, Johannes Zipperle, Daniel Oberladstätter, Sebastian O. Decker, Maik Von der Forst, Kevin Michel Tourelle, Maximilian Dietrich, Markus A. Weigand, Felix C. F. Schmitt

**Affiliations:** 1Department of Anesthesiology, Medical Faculty Heidelberg, Heidelberg University, Im Neuenheimer Feld 420, 69120 Heidelberg, Germany; daniel.gruneberg@med.uni-heidelberg.de (D.G.);; 2Department of Anesthesiology, Kaiserslautern Westpfalz Hospital, 67655 Kaiserslautern, Germany; 3Ludwig Boltzmann Institute for Traumatology, The Research Center in Cooperation with AUVA, 1200 Vienna, Austria; 4Department of Anaesthesiology and Intensive Care Medicine, AUVA Trauma Centre Salzburg, Academic Teaching Hospital of the Paracelsus Medical University, 5020 Salzburg, Austria

**Keywords:** ClotPro^®^, ROTEM^®^, algorithm, comparison, viscoelastic hemostatic assays, PPH

## Abstract

**Background**: Viscoelastic hemostatic assays (VHAs) have become an integral diagnostic tool in guiding hemostatic therapy, offering new opportunities in personalized hemostatic resuscitation. This study aims to assess the interchangeability of ClotPro® and ROTEM® delta in the unique context of parturient women. **Methods**: Blood samples from 217 parturient women were collected at three timepoints. A total of 631 data sets were eligible for our final analysis. The clotting times were analyzed via extrinsic and intrinsic assays, and the clot firmness parameters A5, A10, and MCF were analyzed via extrinsic, intrinsic, and fibrin polymerization assays. In parallel, the standard laboratory coagulation statuses were obtained. Device comparison was assessed using regression and Bland–Altman plots. The best cutoff calculations were used to determine the VHA values corresponding to the established standard laboratory cutoffs. **Results**: The clotting times in the extrinsic and intrinsic assays showed notable differences between the devices, while the extrinsic and intrinsic clot firmness results demonstrated interchangeability. The fibrinogen assays revealed higher values in ClotPro^®^ compared to ROTEM^®^. An ROC analysis identified VHA parameters with high predictive values for coagulopathy exclusion and yet low specificity. **Conclusions**: In the obstetric setting, the ROTEM^®^ and ClotPro^®^ parameters demonstrate a significant variability. Device- and indication-specific transfusion algorithms are essential for the accurate interpretation of measurements and adequate hemostatic therapy.

## 1. Introduction

The use of viscoelastic hemostatic assays (VHAs) for diagnostic use and for the guidance of hemostatic therapy is increasing worldwide, and new technologies continue to emerge [1,2,3,4,5]. Currently, VHAs complete or even replace standard laboratory testing and, therefore, represent a cornerstone of modern hemostatic resuscitation [6,7,8]. Viscoelastic hemostatic assays provide more detailed insights into the complex pathophysiology of bleeding than standard coagulation tests [9]. Further viscoelastic testing allows for the shortening of turnover times, providing a crucial time benefit for the treatment of major bleeding [2]. Point-of-care (POC) devices permit precise targeted therapy and reduce over- and undertreatment with blood products and coagulation factor concentrates [6,8,10]. Beside the technical aspects of the devices, appropriately validated algorithms are fundamental in translating diagnostic findings into appropriate therapeutic decisions. 

Currently, a variety of VHA-based transfusion algorithms exist. These algorithms are based mainly on established devices like ROTEM^®^ and TEG^®^ [11,12,13]. The ClotPro^®^ hemostatic device is a new point-of-care monitor for viscoelastic testing. It shares the basic principles of rotational thrombelastometry with ROTEM^®^ and TEG^®^, but significant differences exist [14,15,16]. In a twin-bearing guidance system, the new device is working with a fixed pin and a rotating cup. Excursion is measured through the highly sensitive capacitive detection of forces acting on the pin [17]. ROTEM^®^ delta, in comparison, is working with a fixed cup and a rotating pin, and the rotation angle is detected optically. Furthermore, differences in the activation reagents and cup geometry exist. While ROTEM^®^ delta uses liquid reagents, ClotPro^®^ implements the active tip technology where dry reagents are placed on a sponge within the pipet tip. Therefore, the number of pipetting steps is reduced, and less dilution of the sample occurs compared to the ROTEM^®^ workflow. Ex-test and EXTEM are both activated by means of a recombinant tissue factor, and intrinsic assays are triggered using ellagic acid. Differences exist in platelet inactivation based on the fibrin polymerization assays used. In the ClotPro^®^ Fib-test, platelet function is inhibited dually with cytochalasin D and tirofiban. In ROTEM^®^ FIBTEM, platelets are inactivated solely by cytochalasin D. All these differences suggest that the results from both test systems cannot be used interchangeably. The first reports on the use of these systems in cardiac surgery, critical ill patients, and pediatric individuals confirm this hypothesis [14,16,18]. They found that the clot firmness values of extrinsically activated tests could be used interchangeably, while fibrin polymerization MCF and clotting times showed significant differences [14,16]. While patients undergoing cardiac surgery and presenting a critical illness are, by nature, more likely to have a compromised coagulation state (iatrogenic or as a result of the underlying disease) parturient women show strong activation of the coagulation system. To date, it is not known to which extent the cutoffs derived using established VHA-based algorithms for PPH are able to be applied to the ClotPro^®^ system. 

We hypothesized that the results from viscoelastic hemostatic measurements depend on the device used and also on special clinical circumstances, and, therefore, VHA-based transfusion algorithms should be specified for both the device used and the particular patient population observed, with their typical hemostatic range. Therefore, this study examined differences in the VHA parameters between the ROTEM^®^ and ClotPro^®^ systems in parturient women. Furthermore, we compared the VHA parameters to standard laboratory coagulation parameters.

## 2. Materials and Methods

To determine the interchangeability of the VHA devices ROTEM^®^ delta and ClotPro^®^, we examined parturient women utilizing a descriptive study design. This study was conducted as a secondary analysis of the FYPPREG study, which is published elsewhere [19]. The underlying study was approved by the Ethics Committee of the Medical Faculty of Heidelberg (trial code no. S-759/2019) and is registered at the German clinical trials’ register (DRKS00021531).

### 2.1. Participants

Parturient women with an age ≥ 18 years who visited Heidelberg University Women’s Hospital between June 2020 and December 2020 and signed their written informed consent were included in this study. In contrast to the FYPPREG study, women with a history of coagulopathy were included as well. 

### 2.2. Measurements

For each patient, blood samples were collected at three different timepoints (hospital admission, 30–60 min after placental separation, and the first day postpartum). At each timepoint, two vials of 3 mL of citrate-anticoagulated whole blood were obtained for POC-VET measurements (ROTEM^®^ delta and ClotPro^®^) and standard laboratory tests. All the blood samples were handled by the same experimenter. The point-of-care viscoelastic-test (POC-VET) measurements were performed immediately after sampling. For the standard laboratory tests, the blood samples were sent to the central laboratory facility via tube mail. The following standard coagulation tests were performed: the prothrombin time (normal range 70–125%), the activated partial thromboplastin time (normal range < 35 s), and the fibrinogen according to the Clauss method (normal range 1.5–3.8 g/L; Clauss fibrinogen assay, analyzed with a Sysmex CS 5100 coagulation analyzer, Sysmex, Kobe, Japan) were measured in the laboratory facility at Heidelberg University Clinic. Regarding the POC-VET analysis, we ran EXTEM, INTEM, and FIBTEM with the ROTEM^®^ delta system and the respective tests for the ClotPro^®^ (Ex-test, In-test, and Fib-test). To determine the platelet contribution, we calculated the extrinsically activated maximum clot elasticity (MCE) minus the fibrin polymerization assay MCE according to Lang et al. [20]. All the measurements were performed in agreement with the respective manufacturer’s guidelines.

### 2.3. Statistical Methods 

Data distribution was tested using the Shapiro–Wilk normality test. Continuous variables were expressed as the mean and 95% limits of agreement (LOA) or 95% confidence limits, as appropriate. The following assays were compared: In-test and INTEM; 2. EX-test and EXTEM; and 3. Fib-test and FIBTEM. For the first two of these pairs of assays, the clotting time (CT), the clot firmness at 5 and 10 min (A5; A10), and the maximal clot firmness (MCF) were compared. For the third pair, the clot firmness at 5 and 10 min (A5; A10) and the maximal clot firmness (MCF) were compared. Correlations between the ClotPro^®^ and ROTEM^®^ delta measurements were calculated using Pearson’s correlations. The correlations were classified as poor (0.1–0.2), fair (0.3–0.5), moderate (0.6–0.7), very strong (0.8–0.9), and perfect (1.0) according to Chan et al. [21]. Fit plots were drawn for the ClotPro^®^ vs. ROTEM^®^ parameters, and Pearson’s correlation coefficient was calculated. A regression line with a 95% CI was shown within the fit plot. To determine the type and magnitude of bias between both devices, Bland–Altman plots were generated. Therefore, the average value between both methods was plotted against the differences between the ClotPro^®^ and ROTEM^®^ parameters. To calculate the relative bias, the differences were divided by the respective average between both methods. Regression lines for the differences between both methods were shown within the BA plot to depict the influence of the measurement range on the magnitude and direction of the bias. For the optimal cutoff calculations, ROC analyses were performed, and the best cutoff was calculated by maximizing the Youden index. The positive predictive value (PPV) was calculated as the number of true positive results/(number of true positive results + number of false positive results). The negative predictive value (NPV) was calculated as the number of true negative results/(number of true negative results + number of false negative results). All the statistical analyses and visuals were created with SAS version 9.4 (SAS Institute Inc., Cary, NC, USA).

## 3. Results

A total of 217 patients met the predefined inclusion criteria. A total of 631 paired measurements contained complete data sets for ROTEM^®^, ClotPro^®^, and standard laboratory tests and were, therefore, eligible for our final analysis. Twenty data sets lacked either ClotPro^®^ or ROTEM^®^ data and were, thus, excluded from our analysis. The demographic data and baseline characteristics of the parturient women are shown in Table 1. 

### 3.1. Tissue Factor-Triggered Viscoelastic Assays

Figure 1 shows the results for the tissue factor-triggered viscoelastic assays (Figure 1).

#### 3.1.1. Clot Initiation

We found a fair correlation of the clotting times in the Ex-test vs. the EXTEM assays. The regression coefficient was 0.34 with a *p*-value < 0.001. The BA plots show a mean difference of −6 s (−14.2%) in the ClotPro^®^ Ex-test CT vs. the ROTEM^®^ EXTEM CT. The 95% LOA ranged from −22 s to +10 s (−47% to +19%). The regression line of the differences shows that the ClotPro^®^ Ex-test CT provides lower values in the lower range of measurements until 60 s and higher values in the upper range of measurements compared to the ROTEM^®^ EXTEM CT. 

#### 3.1.2. Clot Firmness

A5, A10, and MCF all show very strong correlations between the ClotPro^®^ Ex-test and the ROTEM^®^ EXTEM. The correlation coefficients were all > 0.8 (0.85, 0.84, and 0.83 for A5, A10, and MCF, respectively; *p*-values < 0.0001). At the A5 timepoint, ClotPro^®^ showed a bias of +0.8 mm (+1.8%) with a 95% LOA of −5 mm to +7 mm (−10 to +14%). In contrast, the A10 and MCF values showed a negative bias of −2 mm (−3.1%) for A10 and −5 mm (−7.2%) for MCF. The regression lines of the differences indicate that ClotPro^®^ showed higher values in the bottom range of measurements and lower values in the upper end of the measurement range. The relative 95% LOA was smallest for the A10 values (−12.6 to +6.1%). At 61 mm, the A10 amplitude mean bias was zero, but there was still a random bias between both devices.

### 3.2. Ellagic Acid-Triggered Viscoelastic Assays

Figure 2 presents the results for the ellagic acid-triggered viscoelastic assays.

#### 3.2.1. Clot Initiation

The In-test and INTEM assays showed a fair correlation with a regression coefficient of 0.48 (*p* < 0.0001). The BA plots showed a mean difference of −11 s (−7.1%) in the ClotPro^®^ In-test CT vs. the ROTEM^®^ INTEM CT. The 95% LOA ranged from −52 s to +29 s (−32.4% to +18.1%). The regression line of the differences showed that ClotPro^®^ provides higher values compared to ROTEM^®^ in the lower range of measurements (below 135 s) and higher results for values above 135 s.

#### 3.2.2. Clot Firmness

Regarding clot firmness, the comparison between ClotPro^®^ and ROTEM^®^ showed very strong correlations for A5, A10, and MCF. The correlation coefficients were >0.9 for A5 and A10 and 0.87 for MCF. The Bland–Altman plots revealed that the ClotPro^®^ measurements of clot firmness were slightly lower compared to the ROTEM^®^ measurements at all three timepoints. The greatest relative bias was seen for the A10 values with a mean difference of −4 mm (−7.2%). The smallest relative 95% LOA was found for MCF (−8 mm to 0 mm; −13.4% to −0.3%). The regression lines for the differences showed that the In-test clot firmness in ClotPro^®^ was lower for values above 40 mm, while amplitudes in the bottom range of measurements were higher compared to the ROTEM^®^ clot firmness measurements. 

### 3.3. Fibrin Polymerization Assays

Figure 3 shows the results for the fibrin polymerization assays.

#### Clot Firmness

The regression plots show a very strong correlation between the ClotPro^®^ Fib-test and the ROTEM^®^ FIBTEM clot firmness. The correlation coefficient was ≥ 0.85 for all three parameters. The *p* values were < 0.0001, respectively. The strongest correlation between the Fib-test vs. the FIBTEM was found for MCF.

The Bland–Altman plots showed higher values for the Fib-test clot firmness parameters A5, A10, and MCF compared to the corresponding ROTEM^®^ FIBTEM values. The relative mean bias was greatest for MCF +16% with a 95% LOA ranging from −2.8% to +32.8%. Likewise, a relevant bias was found for the A10 values with a 95% LOA ranging from −5.6% to +29.2%. The regression lines of the differences revealed that positive bias was greatest in the bottom range of measurements, while values in the upper range of measurements showed less differences but were still higher in ClotPro^®^ compared to ROTEM^®^.

### 3.4. AUC Analysis and Best-Cutoff Calculations

ROC analyses were performed to determine the agreement between the VHA parameters and standard laboratory tests (Table 2 and Figure 4). We found good agreement between the Ex-test/EXTEM CT, the Fib-test/FIBTEM A10, and the platelet contribution and their corresponding standard laboratory parameters, i.e., prothrombin time (PT), fibrinogen, and platelets. The calculated ClotPro^®^ cutoffs for the Ex-test CT and the Fib-test A10 had a higher sensitivity compared to the respective ROTEM^®^ parameters, while specificity was similar for both devices. Regarding platelet contribution, ROTEM^®^ showed a higher sensitivity but a lower specificity compared to the ClotPro^®^ cutoffs. For both devices, the calculated VHA cutoffs provided good negative predictive values, while the positive predictive values were low.

Table 3 gives an additional overview of the main findings of the manuscript (Table 3).

## 4. Discussion

In a secondary analysis of 217 patients resulting in 631 paired measurements, we found that the clotting times in extrinsic and intrinsic assays showed remarkable differences between the ClotPro^®^ and ROTEM^®^ delta devices, whereas the extrinsic and intrinsic clot firmness measurements could be used interchangeably. The ClotPro^®^ Fib-test clot firmness showed systemically higher values compared to the ROTEM^®^ FIBTEM measurements. Especially in situations of low fibrinogen concentration, ClotPro^®^ provides higher clot firmness values compared to ROTEM^®^ delta. Our results confirm existing data from Yoshii et al. and Infanger et al. where the clot firmness values in intrinsic and extrinsic assays can be used interchangeably, while the clot firmness measured in a fibrin polymerization assay cannot [14,16]. Even though these differences between the two systems and the fibrin polymerization tests appear to be the most surprising results of this study, they are in line with the daily clinical experience of using these two different systems. In our study, the Fib-test clot firmness parameters were constantly higher compared to their FIBTEM equivalents over a broad range of measurements. This is of particular interest as the Fib-test applied the dual inhibition of platelets, meaning that its measured clot amplitude should be lower compared to the FIBTEM [22,23]. Schlimp et al. clearly demonstrated that the dual inhibition of platelets provides an almost complete elimination of platelet contribution to clot firmness. Therefore, one of the slight technical differences between the systems (cup-rotating ClotPro^®^ vs. pin-rotating ROTEM^®^) or a difference in the detection software used might be the reason for the higher values in the Fib-test clot firmness. Even though it was beyond the scope of the present study to investigate the particular reason for this deviation, it could constitute a relevant difference in clinical practice and should be kept in mind. From the authors’ point of view, this is the most important impact of this study on the current management of PPH. While a ROTEM^®^ FIBTEM A10 of 20 mm corresponds to a fibrinogen concentration of 4 g/L, a ClotPro^®^ Fib-test A10 of 23 mm is necessary to reach the same plasma fibrinogen concentration. Since fibrinogen is one of the first coagulation factors consumed in active bleeding, timely fibrinogen substitution and avoidance of hypofibrinogenemia are key factors in the successful management of PPH.

The current study also demonstrates a strong-to-very-strong correlation of clot amplitude between the different devices in the extrinsic and intrinsic assays among parturient women. The clot firmness parameters for the Ex- and In-test showed only slight differences. From our point of view, the observations presented in this study explain the conflicting results arising from previous investigations. Our data revealed lower values of clot firmness in the Ex-test A10, with a mean bias of −2.0 mm, compared to the ROTEM^®^ delta EXTEM A10, while Yoshii et al. found higher values for ClotPro^®^, with a mean bias of +2.4 mm [16]. The reason for this is most probably based on the different patient populations enrolled in the two studies. The coagulation system in parturient women is maximally stimulated towards a pro-coagulant response, while this is not the case in patients undergoing CPB surgery. In the latter, patients suffer from iatrogenic (heparin, CPB, hypothermia) and surgical impacts (blood loss, tissue trauma) on their coagulation system. Therefore, Yoshii et al. examined VHAs in a range of 30–60 mm Ex-test A10, while the patients in our study presented with Ex-test A10 values between 50 and 70 mm. These findings have implications for clinical practice, particularly in showing that specific reference ranges for designated patient groups might be defined to facilitate patient-centered individualized hemostatic resuscitation. 

The clotting times in ClotPro^®^ and ROTEM^®^ showed weak correlations between the two devices and standard laboratory tests. The clotting times in ClotPro^®^ showed lower values in the lower end of the measurement range for the Ex-test and higher values in the lower end of the measurements for the In-test. Although the clotting time (CT) is the first measurement result available, it is influenced not only by the activity of coagulation factors and anticoagulants such as heparin, vitamin-K-antagonists, or DOACs but also to a large extent by the concentration of available fibrinogen [24]. Therefore, for a correct assessment of CT, the simultaneous measurement of fibrin polymerization (FIBTEM/Fib-test) is necessary. Only when the FIBTEM/Fib-test clot amplitude is normal and the CT value remains significantly prolonged can this be considered an indication of restricted thrombin generation. Whether the high susceptibility of external confounders could be an explanation of weak correlation is questionable, especially in simultaneously measured patient samples. Therefore, only differences in the mode of detection can be a reason behind said weak correlation. 

ROTEM^®^ delta is an established viscoelastic test system that had been used for many years in clinical practice [25]. Existing transfusion algorithms provide cutoffs for ROTEM^®^ parameters to guide the substitution of blood products and coagulation factor concentrates [26,27]. ClotPro^®^ is a relatively new device that is increasingly being used in Europe and worldwide [16,18,28,29,30,31]. In current clinical practice, ClotPro^®^ viscoelastic hemostatic assays have been used, but therapeutic consequences continue to be deduced from established algorithms based on ROTEM^®^ or TEG parameters [11,12,32,33]. However, data from cardiac surgery and critically ill COVID-19 patients have suggested that ROTEM^®^ and ClotPro^®^ parameters cannot be used interchangeably. Differences in pin and cup geometry, materials, chemicals, and detection technologies may lead to different kinetics and absolute values in clot formation times and clot firmness. With the present study we provide evidence of the need for transfusion algorithms that are specific to the VHA device being used and fitted to the distinct clinical setting in which they are meant to be used. Because of the complex relation between ClotPro and ROTEM test results, it is not sufficient to define a single correction value for converting ClotPro values into ROTEM algorithms. In clinical practice, it is necessary to use a specific transfusion algorithm that is based on ClotPro data. Using a ROTEM-based transfusion algorithm may lead to over- or undertreatment depending on the specific range of measurements. For comparative studies between devices, it is important to interpret the results in the context of a measurement range, because the generalization of results over a broad range of measurements could lead to equalizing existing differences. VHAs are shown to facilitate coagulation therapy due to their faster turnover times and higher sensitivity for early signs of coagulopathy [7,34,35]. An improved mortality rate has been shown thanks to the use of VHAs compared to standard laboratory tests [36]. VHAs’ benefits arise from their short turnover times and the extended test panels, able to answer specific clinical questions. Nevertheless, the cutoff values in established transfusion algorithms often lack robust evidence. In the future, it is necessary to further increase the sensitivity and specificity of VHAs to reduce overtreatment and facilitate faster response times when coagulopathy emerges. With the increasing use of viscoelastic hemostatic testing, new devices have been developed, and the results differ between analyzers. The present results suggest that device- and field-specific algorithms are crucial in improving patient-centered hemostatic therapy and, potentially, the outcomes and in lowering costs. 

### Limitations

The main focus of this manuscript was the comparison of two POC-VET systems, ClotPro^®^ and ROTEM^®^ delta. The present study was conducted as a secondary analysis of the FYPPREG study, which evaluated the significance of pre-partum TPA-test lysis time in the prediction of PPH. Due to the study population, a high number of patients showed a significantly activated coagulation system in the third trimester, which makes it difficult to compare our data with studies from other settings like trauma or cardiac surgery settings. Another limitation of our study arises from the fact that, in the present data set, the proportion of patients with severely compromised coagulation parameters was low. 

## 5. Conclusions

In the present day, the established algorithms for hemostatic resuscitation are based on data deriving from ROTEM^®^ or TEG^®^. However, existing data from head-to-head comparison between ClotPro^®^ and ROTEM^®^ indicate that the reference values are not interchangeable between the devices. Up until now, no data were available for an obstetric setting. Therefore, we performed a head-to-head comparison of the Haemonetics ClotPro^®^ and ROTEM^®^ delta devices. Utilizing 631 paired measurements during childbirth, we showed that the extrinsic and intrinsic clot firmness parameters can be used interchangeably between devices, while clot firmness in fibrinogen polymerization assays as well as ex- and intrinsic clotting times show remarkable differences. We revealed that the differences between test results are complex and depend on the range of measurements. Therefore, device-specific treatment algorithms need to be established, rather than converting ClotPro^®^ results into ROTEM^®^ algorithms. The authors conclude that device- and indication-specific transfusion algorithms are needed to improve sensitivity and specificity for the detection of coagulopathy, improve patient outcome, and lower costs.

## Figures and Tables

**Figure 1 jcm-13-00692-f001:**
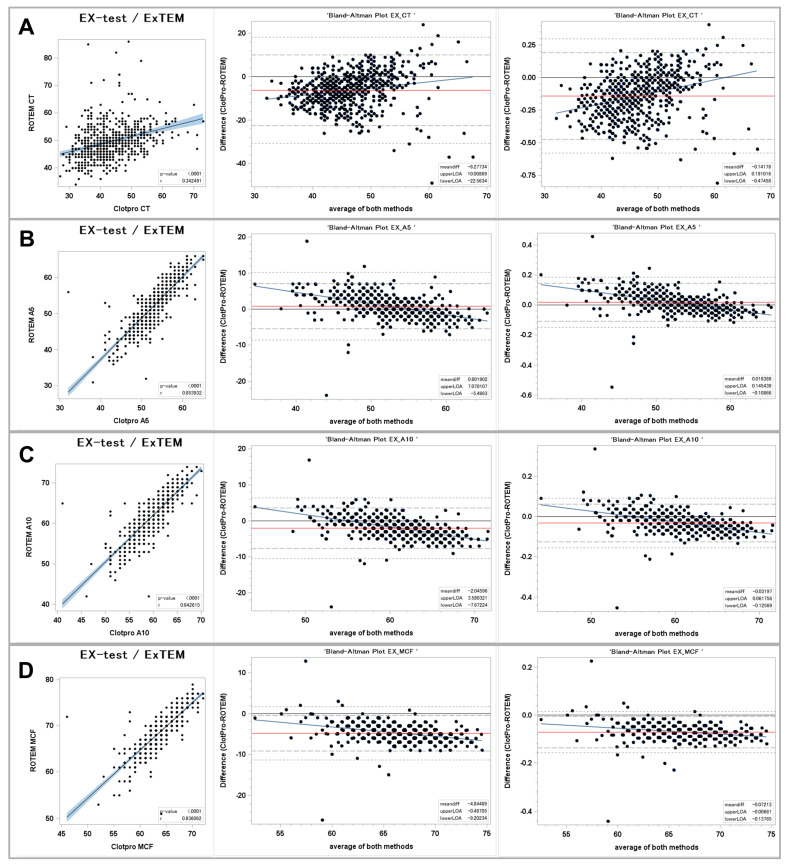
Comparison of extrinsically stimulated assays between ClotPro^®^ and ROTEM^®^. (**A**): clotting time; (**B**): clot firmness after 5 min (A5 value); and (**C**): clot firmness after 10 min (A10 value); (**D**): maximal clot firmness (MCF value). On the left: Fit plot with a regression line. The blue area around the regression line represents 95% confidence limits for the regression line. The *p*-value and Pearson’s correlation coefficient “r” are in the lower right corner. In the middle: Bland–Altman plot of the absolute values. On the right: Bland–Altman plot of the relative values. The solid black line indicates zero difference; the red solid line represents the actual mean bias between the ClotPro^®^ and ROTEM^®^ parameters; the dashed lines represent the 95% limits of acceptance (95% LOA); the dotted line represents the 99% LOA; and the blue line represents the regression line for the difference between ClotPro^®^ and ROTEM^®^. The numeric values of the mean bias and of the lower and upper 95% LOA are given in the bottom right corner. The clotting times are measured in seconds, and the clot firmness is measured in mm amplitude, respectively.

**Figure 2 jcm-13-00692-f002:**
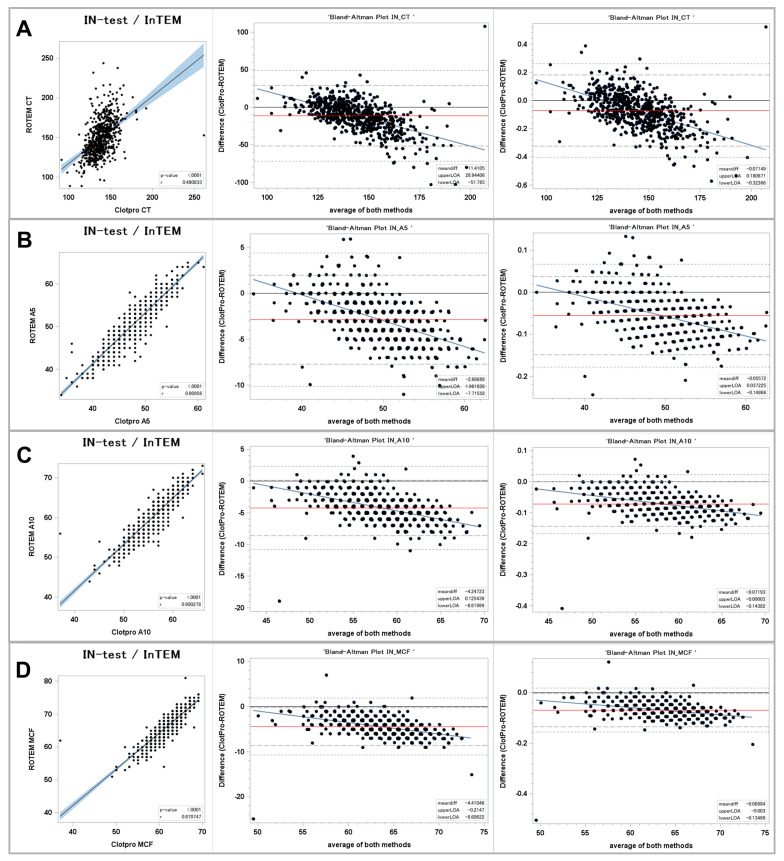
Comparison of intrinsically stimulated assays between ClotPro^®^ and ROTEM^®^. (**A**): clotting time; (**B**): clot firmness after 5 min (A5 value); (**C**): clot firmness after 10 min (A10 value); and (**D**): maximal clot firmness (MCF value). On the left: Fit plot with a regression line. The blue area around the regression line represents the 95% confidence limits for the regression line. The *p*-value and Pearson’s correlation coefficient “r” are in the lower right corner. In the middle: Bland–Altman plot of the absolute values. On the right: Bland–Altman plot of the relative values. The solid black line indicates zero difference; the red solid line represents the actual mean bias between the ClotPro^®^ and ROTEM^®^ parameters; The dashed lines represents the 95% limits of acceptance (95% LOA); the dotted line represents the 99% LOA; and the blue line represents the regression line for the difference between ClotPro^®^ and ROTEM^®^. The numeric values of the mean bias and of the lower and upper 95% LOA are given in the bottom right corner. The clotting times are measured in seconds, and the clot firmness is measured in mm amplitude, respectively.

**Figure 3 jcm-13-00692-f003:**
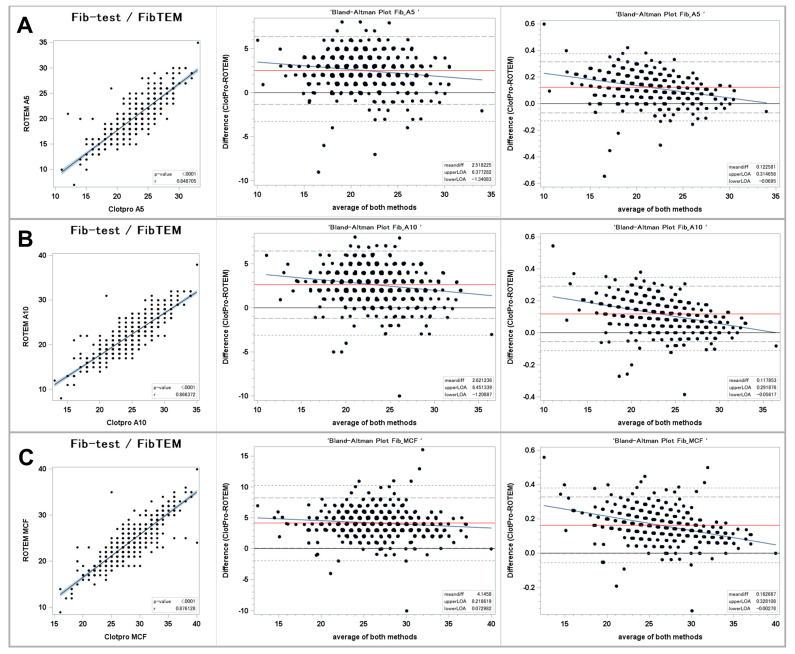
Comparison of fibrin polymerization assays between ClotPro^®^ and ROTEM^®^. (**A**): clot firmness after 5 min (A5 value); (**B**): clot firmness after 10 min (A10 value); and (**C**): maximal clot firmness (MCF value). On the left: Fit plot with a regression line. The blue area around the regression line represents the 95% confidence limits for the regression line. The *p*-value and Pearson’s correlation coefficient “r” are located in the lower right corner. In the middle: Bland–Altman plot of the absolute values. On the right: Bland–Altman plot of the relative values. The solid black line indicates zero difference; the red solid line represents the actual mean bias between the ClotPro^®^ and ROTEM^®^ parameters; the dashed lines represents the 95% limits of acceptance (95% LOA); the dotted line represents the 99% LOA; and the blue line represents the regression line for the difference between ClotPro^®^ and ROTEM^®^. The numeric values of the mean bias and of the lower and upper 95% LOA are given in the bottom right corner. The clotting times are measured in seconds, and the clot firmness is measured in mm amplitude, respectively.

**Figure 4 jcm-13-00692-f004:**
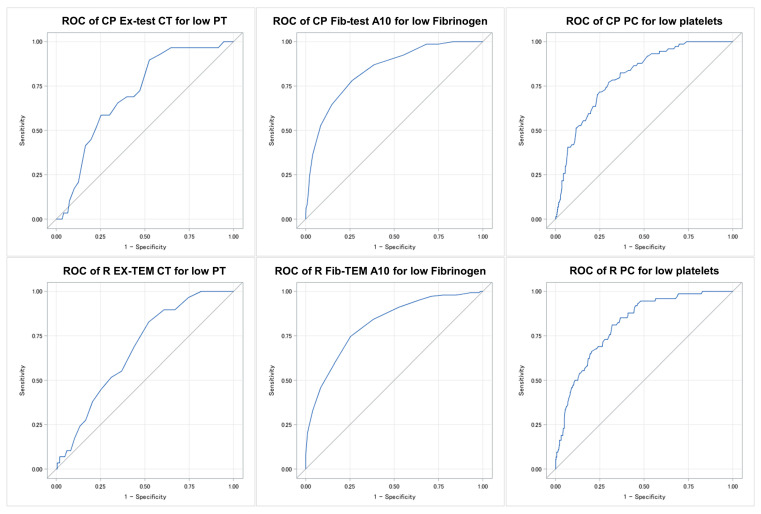
Receiver operating characteristics (ROC) curve of the VAH parameters for defined laboratory cutoffs. Low PT = PT < 100 s; low fibrinogen = fibrinogen < 4 g/L; low platelets = platelet count < 150/nL; CP = ClotPro^®^; R = ROTEM^®^; PT = prothrombin time and PC = platelet contribution.

**Table 1 jcm-13-00692-t001:** Baseline characteristics of the women analyzed in the present study.

	Total n = 217
Age [y]	34 (30–37)
Gestation age [wk]	39 (38–40)
Gemini	10 (4.6%)
Gravidity	2 (1–3)
Parity	1 (0–1)
Nullipara	104 (47.9%)
Prior abortions	0 (0–1)
Spontaneous birth [n]	109 (50.2%)
Primary cesarian [n]	70 (32.3%)
Secondary cesarian [n]	38 (17.5%)
Uterine atony	4 (1.8%)
Coagulopathy (all kinds)	11 (5.1%)
vWD	3 (1.4%)
Factor V Leiden mutation	4 (1.8%)
Factor XII deficiency	2 (0.9%)
HELLP/preeclampsia	5 (2.3%)
Birth injury	87 (40.1%)

The values are given as the median (Q1–Q3) or as a number and percentage, as appropriate. vWD = von Willebrand disease; coagulopathy (all kinds) = number of individuals who presented at least one of the below-listed coagulation disorders.

**Table 2 jcm-13-00692-t002:** Cutoff calculations for ROTEM^®^ and ClotPro^®^ parameters based on standard laboratory values.

	N	AUC (95% CL)	Best Cutoff	Youden Index	Sens.	Spec.	PPV	NPV
PT < 100% vs. ClotPro^®^ Ex-test CT [s]	29 vs. 634	0.7074 (0.63–0.79)	42 s	0.37	89.7%	47.5%	0.08	0.99
PT < 100% vs. ROTEM^®^ EXTEM CT [s]	29 vs. 602	0.6748 (0.59–0.76)	49 s	0.31	82.8%	47.8%	0.07	0.93
Fibrinogen < 4 g/L vs. ClotPro^®^ Fib-test A10 [mm]	146 vs. 485	0.8387 (0.80–0.87)	23 mm	0.52	78.1%	73.8%	0.47	0.92
Fibrinogen < 4 g/L vs. ROTEM^®^ FIBTEM A10 [mm]	146 vs. 485	0.8164 (0.78–0.86)	20 mm	0.49	74.6%	74.6%	0.56	0.87
Platelets < 150/nL vs. ClotPro^®^ Platelet contribution [mm]	74 vs. 557	0.8023 (0.75–0.85)	135 mm	0.47	77.0%	70.0%	0.28	0.96
Platelets < 150/nL vs. ROTEM^®^ Platelet contribution [mm]	74 vs. 557	0.8140 (0.77–0.86)	183 mm	0.49	81.1%	68.0%	0.13	1.00

PT = prothrombin time; ClotPro^®^ platelet contribution = Ex-test MCE—Fib-test MCE; ROTEM^®^ platelet contribution = EXTEM MCE—FIBTEM MCE; AUC = area under the curve; Sens. = sensitivity; Spec. = specificity; PPV = positive predictive value; and NPV = negative predictive value.

**Table 3 jcm-13-00692-t003:** Summary of the main findings of the manuscript.

ClotPro^®^ Parameters	Mean Difference to Respective ROTEM^®^ Parameter
Ex-test CT	↓ (−6 s)
In-test CT	↓ (−11 s)
Ex-test A10	↓ (−2 mm)
Fib-test A10	↑ (+3 mm)

↓ = ClotPro^®^ results lower compared to ROTEM^®^ values; ↑ = ClotPro^®^ results higher compared to ROTEM^®^ values.

## Data Availability

Data are contained within the article.

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
