# Peer review of "Comparison of Two Viscoelastic Testing Devices in a Parturient Population"

_jcm, 2024, doi:10.3390/jcm13030692_

Round 1
Reviewer 1 Report
Comments and Suggestions for Authors|
No |
Present |
Notes |
|
74 |
Statistical methods |
Authors need to clarify the study design, it is cross section, or descriptive or … |
|
Table no 1 |
Age |
Age of the patient in patient and method from 18 years but the minimal age is 30 ??? |
|
Table no 1 |
Coagulopathy |
Which type of coagulopathies |
|
Table no 2 |
AUC |
The authors may need to add figure of ROC correlation to the manuscript |
|
Table no 2 |
PPV = positive predictive value |
The authors need to add way of PPV calculation |
|
321 |
Conclusion |
To be summarize to more focus way |
|
|
|
|
Author Response
We would like to thank you for taking your time to review our manuscript critically. We are glad to get the opportunity to submit a revised version of our manuscript titled “Comparison of two viscoelastic testing devices in a parturient population” to JCM. We appreciate your time and effort providing valuable feedback on our manuscript. We were able to incorporate changes to reflect all of your suggestions. Your review was detailed and always constructive. We responded to all of your comments as good as we could and revised the manuscript respectively.
A detailed point to point review is attached as word document.

Reviewer 2 Report
Comments and Suggestions for Authors
Reviewer’s comments
1. This study compared the use of ClotPro and ROTEM. Is ROTEM considered a gold standard as viscoelastic hemostatic assays?
2. What were the other inclusion (except age>18) and exclusion criteria?
3. Is there any reason why blood was taken during admission before delivery, immediately postpartum, and after one day of delivery? Why not during booking or antenatal follow-up?
4. There is a total of 217 patients but the total number of tests performed is only 631 and not 651?
5. Can the author elaborate more regarding the conclusion made at the end of the study? We revealed that comparability is dependent on the range of measurement and therefore on the specific patient group which is investigated.
6. What is the significant impact of this study on the current management of PPH especially those with coagulopathy?
Comments on the Quality of English LanguageAuthor Response
We like to thank you for taking your time to review our manuscript critically. Your review was detailed and always constructive. We responded to all of your comments as good as we could and revised the manuscript respectively. With your helpful comments you definitely improve our work.
A detailed point to point review is attached as word document.
